# *MEK6* Overexpression Exacerbates Fat Accumulation and Inflammatory Cytokines in High-Fat Diet-Induced Obesity

**DOI:** 10.3390/ijms222413559

**Published:** 2021-12-17

**Authors:** Suyeon Lee, Myoungsook Lee

**Affiliations:** 1Department of Food & Nutrition, Sungshin Women’s University, Seoul 01133, Korea; lsy4258@naver.com; 2Research Institute of Obesity Sciences, Sungshin Women’s University, Seoul 01133, Korea

**Keywords:** obesity, *MEK6(MAP2K6)*, p38, inflammatory cytokines, thermogenesis

## Abstract

Obesity is a state of abnormal fat accumulation caused by an energy imbalance potentially caused by changes in multiple factors. *MEK6* engages in cell growth, such as inflammation and apoptosis, as one of the MAPK signaling pathways. The *MEK6* gene was found to be related to *RMR*, a gene associated with obesity. Because only a few studies have investigated the correlation between *MEK6* and obesity or the relevant mechanisms, we conducted an experiment using a Tg*^MEK6^* model with *MEK6* overexpression with non-Tg and chow diet as the control to determine changes in lipid metabolism in plasma, liver, and adipose tissue after a 15-week high-fat diet (HFD). *MEK6* overexpression in the Tg*^MEK6^* model significantly increased body weight and plasma triglyceride and total cholesterol levels. p38 activity declined in the liver and adipose tissues and lowered lipolysis, oxidation, and thermogenesis levels, contributing to decreased energy consumption. In the liver, lipid formation and accumulation increased, and in adipose, adipogenesis and hypertrophy increased. The adiponectin/leptin ratio significantly declined in plasma and adipose tissue of the Tg*^MEK6^* group following *MEK6* expression and the HFD, indicating the role of *MEK6* expression in adipokine regulation. Plasma and bone-marrow-derived macrophages (BMDM) of the Tg*^MEK6^* group increased *MEK6* expression-dependent secretion of pro-inflammatory cytokines but decreased levels of anti-inflammatory cytokines, further exacerbating the results exhibited by the diet-induced obesity group. In conclusion, this study demonstrated the synergistic effect of *MEK6* with HFD in fat accumulation by significantly inhibiting the mechanisms of lipolysis in the adipose and M2 associated cytokines secretion in the BMDM.

## 1. Introduction

Obesity is a state of abnormal fat accumulation resulting from converting excess energy in the body to fat. It is caused by an energy imbalance caused by changes in various complex factors from hormones to dietary or lifestyle habits [1,2,3]. Although the World Health Organization has designated obesity as a disease, a major public health issue that requires prevention and management, the prevalence of obesity has rapidly increased worldwide [4]. Obesity is a risk factor for increased mortality because of diseases, such as hypertension, hypocholesteremia, diabetes, cardiovascular disease, and cancer [5,6,7,8]. Recently, studies have reported obesity as a factor increasing the risk of severe pneumonia and death in COVID-19 patients [9,10]. The pharmacological interventions for obesity treatment mainly focus on inhibiting the absorption and formation of lipids and controlling appetite [11], which has potential safety problems and side effects [12]. In this context, more active R&D should proceed with a focus on restoring energy balance [13]. At the same time, a radical direction should be suggested for personalized obesity treatment based on identifying different markers in each individual on cellular and molecular levels with relevant gene expression.

In a previous pilot GWAS study conducted on pediatric obesity patients, DNA screening was applied to identify the SNPs (Single nucleotide polymorphism, rs9916229, rs756942) on *mitogen-activated protein kinase, kinase 6 (MAP2K6*, *MEK6*), which are correlated with RMR and obesity prevalence [14]. MAPK is responsible for transferring the signals activated by growth hormones, cytokines, and stress receptors from the cell membrane to the nucleus inside the cell, regulating various cell functions, including proliferation, differentiation, and apoptosis [15]. The RNA expression of *MEK6* gene differs depending on organs such as liver, heart, and fat, but most of it is expressed, and *MEK6* is a key component of the MAPK signal transduction pathway with a known role in the regulatory mechanisms for cytokines and stress-induced apoptosis, and various types of inflammatory tumorigenesis [16,17,18,19]. MEK3/6 are upstream factors of p38 as part of the well-known MAPK signaling pathway [20]. It is known that MEK3 activates certain parts of p38 (α, γ, δ) and MEK6 activates most of the p38 (α, β2, γ, δ) [21]. The resulting p38 activity controls uncoupling protein-1 (UCP1) expression in white adipocytes (WATs) to participate in various processes, including inflammation, and cell death, growth, and proliferation [22]. Thus, the T3 treatment of *MEK6* knock-out (KO) WATs was shown to increase the levels of p38α and UCP1 to promote energy production and reduce fat accumulation [23]. In T3-induced beiging (3T3-L1 cell), *MEK6* expression declines simultaneously with adipogenic factors and energy consumption [24].

Regulation of adipogenesis is a strong candidate as a potential strategy for obesity prevention, and adipocyte differentiation plays an essential role in the process of body fat mass [25]. Excessive adipogenesis in the body, especially in WAT, has a risk of inducing obesity and increasing the risk of serious diseases [26]. Cytokines are also produced and secreted by macrophages and adipocytes, which accompany fat accumulation [27]. Indeed, compared to the body weight loss group, the group of obese women exhibited a high level of *MEK6* expression [28], and in 3T3-L1, the MEK6 pathway promoted the induction of insulin resistance by tumor necrosis factor-α (TNF-α) [29] and the process of adipogenesis [30]. In light of this, a close association is suggested between the MEK6 pathway and the mechanisms of energy balance during obesity.

Various studies have been conducted regarding *MEK6*-related cell signaling pathways, but very few have investigated the association with obesity and none on the relevant mechanisms, such as fat accumulation. In this study, we determined whether obesity is caused by *MEK6* overexpression and the mechanisms related to obesity risk factors. Thus, a novel *MEK6* overexpression mouse model (Tg*^MEK6^*) was produced, and the mechanisms behind the synergistic effect with a high-fat diet (HFD) in obesity development were investigated.

## 2. Results

### 2.1. MEK6 Overexpression in Tissues of Tg^MEK6^ Compared to Non-Tg

The MEK6 overexpression for Tg^MEK6^/lean and Non-Tg/lean was determined by mRNA and protein levels. The difference in mRNA expression was measured using RT-PCR of the liver, heart, and WATs, with a MEK6 vector-specific primer, applied in transfection (Figure 1A). The results of MEK6 mRNA expression showed that the expression was approximately 30-fold higher in the Tg^MEK6^/lean compared to Non-Tg/lean, and the expression was also high in Tg^MEK6^/DIO for the liver and eWATs compared to the Non-Tg/DIO (Figure 1B). Western blot was performed for MEK6 protein expression in the liver, eWATs, and BMDM, and the expression in the liver and eWATs was statistically higher by approximately 1.5% higher in the Tg^MEK6^/lean and Tg^MEK6^/DIO than in the Non-Tg/lean and Non-Tg/DIO (Figure 1C).

### 2.2. MEK6 Overexpression on Changes of Weight Gain and Dietary Intake

After MEK6 overexpression, the final body weight, body weight gain, dietary intake, and FER for 15 weeks were measured and compared with the control (Non-Tg) (Table 1). The final body weight was significantly increased in the following order: Tg^MEK6^/DIO, Non-Tg/DIO, Tg^MEK6^/lean, and Non-Tg/lean. Body weight showed an identical trend in increase, indicating a synergistic effect of MEK6 with DIO. The dietary intake was higher for the lean groups than the DIO groups. For the lean groups, Tg^MEK6^/lean showed higher dietary intake than the Non-Tg/lean. However, for the DIO groups, no influence of gene was found between the Non-Tg/DIO and Tg^MEK6^/DIO. FER was also not different between the Tg^MEK6^/lean and Non-Tg/lean. The heart and kidney weights were significantly higher in the Non-Tg/lean and Tg^MEK6^/lean, whereas the liver and total WAT (pWAT, vWAT, and eWAT) were higher in the Non-Tg/DIO and Tg^MEK6^/DIO because of the HFD.

### 2.3. MEK6 Overexpression Alters Plasma Biochemistry

AST and ALT were measured for the hepatotoxicity test. The levels were significantly higher in the Tg^MEK6^/lean than Non-Tg/lean and in the DIO groups than in the lean groups, whereas the plasma AST and ALT levels were increased within 2–4 times above the ULN (upper limit of normal) [31], indicating the lack of hepatotoxicity because of MEK6 transfection or HFD (Table 2). The AST/ALT ratio was significantly reduced because of MEK6 transfection or HFD. The blood glucose level did not vary between the Tg^MEK6^/lean and Non-Tg/lean, although a significant increase because of HFD was observed. The plasma TG and TC had significantly high values according to MEK6 transfection and HFD; in the case of plasma TC, the DIO groups had approximately 1.5-fold higher values than the lean groups. Thus, assuming that no hepatotoxicity was induced, MEK6 appeared to have increased plasma lipids in relation to HFD.

### 2.4. MEK6 Overexpression on Activation of p38/ERK Associated with Thermogenesis in WAT

MEK6 overexpression had an effect to significantly reduce the protein expression related to p38/ERK activity, thermogenesis (UCP1), and lipolysis (pHSL, HSL) in eWATs. The pERK/ERK expression was significantly lower in the Tg^MEK6^/lean than in the Non-Tg/lean and even lower in the DIO groups, and pp38/p38 expression exhibited a decreasing MEK6 expression-dependent trend (Figure 2A). The activity of HSL, an enzyme in lipolysis, was significantly lower in the Tg^MEK6^/lean than in the Non-Tg/lean and even lower in the DIO groups (Figure 2B). The UCP1, a thermogenesis marker, also showed a significantly decreasing trend similar to HSL (Figure 2C). Overall, MEK6 overexpression appeared to suppress the expression of genes related to lipolysis and energy consumption through the downregulation of the p38 and ERK activities in eWATs.

### 2.5. MEK6 Overexpression Induces Adipocyte Hypertrophy and Adipogenesis in the WAT

To verify whether adipogenesis in WATs was controlled in mice with MEK6 overexpression, H&E staining and western blot analysis were performed. The size of the stained adipocytes in sWATs showed a uniform increase in the Tg^MEK6^/lean compared to the Non-Tg/lean in accordance with MEK6 expression. The DIO groups exhibited adipocyte hypertrophy (Figure 3A). Analyzing the adipocyte size using ImageJ software confirmed a significant increase in the following order: Non-Tg/lean < Tg^MEK6^/lean < Non-Tg/DIO < Tg^MEK6^/DIO (Figure 3B). The increase in adipocyte hypertrophy was accompanied by a significant increase in the expressions of C/EBPα, PPARγ, and SREBP1, the main transcription factors regulating adipocyte differentiation in eWATs (Figure 3C). Overall, increased adipocyte size was induced when the MEK6 expression promoted adipocyte differentiation in WATs, with HFD playing a synergistic role.

### 2.6. MEK6 Overexpression Regulates PPARα, UCP2, and Activation of p38/ERK in the Liver

In the liver, correlations occurred among the expression of p38, ERK, PPARα, and UCP2 related to MEK6 and the MAPK family. Although pERK/ERK expression was higher in the Tg^MEK6^/lean than in the Non-Tg/lean, pp38/p38 expression was significantly lower in Tg^MEK6^/lean, as in the case of eWATs (Figure 4A). Thus, ERK activity in the liver increased MEK6 expression and significantly reduced the phosphorylation of p38. Additionally, PPARα and UCP2 protein expression were examined because the two genes are known to engage in energy consumption contributing to lipolysis activation in the liver. The results showed that HFD had increased expression of both PPARα and UCP2; PPARα did not significantly vary between the Tg^MEK6^/DIO and Non-Tg/DIO. UCP2 was significantly reduced in the Tg^MEK6^/lean and Tg^MEK6^/DIO compared to the Non-Tg/lean and Non-Tg/DIO (Figure 4B). Overall, the MEK6 overexpression appears to cause a decline in UCP2 expression by inhibiting p38 activity to play a role in energy metabolism in the liver.

### 2.7. MEK6 Overexpression Regulates Fat Accumulation and Lipogenic Factors in the Liver

For the morphological analysis of the effects of MEK6 overexpression on lipid metabolism in the liver, H&E staining was used for the histopathological examination of the liver tissue. There was a severe state of lipid accumulation in the liver in the Tg^MEK6^/DIO compared to the Non-Tg/DIO (Figure 5A). In the kidney, the glomerulus exhibited more intense hypertrophy in the Tg^MEK6^/DIO compared to the Non-Tg/DIO (data not shown). Histological changes in the liver were examined based on TG and TC content. Both TG and TC increased based on MEK6 transfection and DIO in the following order: Non-Tg/lean < Tg^MEK6^/lean < Non-Tg/DIO < Tg^MEK6^/DIO. For TG, there was an increase by approximately 1.5–2 times in the Tg^MEK6^/lean and Tg^MEK6^/DIO compared to Non-Tg/lean and Non-Tg/DIO (Figure 5B). The evidence for lipid accumulation in the liver was obtained from the adipogenesis proteins, and a significant increase occurred for PPARγ and SREBP1 engaged in adipogenesis (Figure 5C).

### 2.8. MEK6 Overexpression Regulates Adipokines in the WAT and Plasma

Both adipokines and cytokines are key biomarkers of obesity and insulin resistance. The measured levels of adiponectin and leptin, the adipokines secreted in the adipose tissue, showed that the adiponectin level was lower in the Tg^MEK6^/lean. However, the leptin level was higher, which indicated a significant declining trend in the adiponectin/leptin ratio (Figure 6A). The changes in plasma levels of adiponectin, leptin, TNFα, and IL-10 showed that the level of plasma adiponectin was lower in the Tg^MEK6^/lean than in the Non-Tg/lean, but the level of plasma leptin was higher, which indicated a decreased adiponectin/leptin ratio and hence a correlation with MEK6 expression (Figure 6B). Plasma IL-10 was reduced in the Tg^MEK6^/lean, but no difference was induced for TNFα by MEK6 expression, and the IL-10/TNFα ratio exhibited a significant decrease with MEK6 expression.

### 2.9. MEK6 Overexpression Regulates Cytokines in Bone-Marrow-Derived Macrophage

To examine the characteristics of MEK6 overexpression in primary BMDM, the cells were treated with LPS and IL-4 to induce the differentiation into M1 or M2 macrophage, respectively. The LPS and IL-4 concentrations (50 ng/mL and 20 ng/mL) were determined based on previous studies [32,33], and the macrophage markers were analyzed using western blotting and ELISA. The pro-inflammatory (COX-2, CD86, IL-6) marker expressions in M1 (BMDM stimulated to LPS) exhibited significantly increased COX-2 and IL-6 in the lean groups. The levels were higher in the Tg^MEK6^/lean than in the Non-Tg/lean, and the DIO groups, the levels were higher in the Tg^MEK6^/DIO than in the Non-Tg/DIO (Figure 7A). In contrast, the anti-inflammatory (Arginase-1, C137L, IL-6) marker expressions in M2 (BMDM stimulated to IL-4) showed a significant decrease in the Tg^MEK6^ groups. The levels were lower in the Tg^MEK6^/lean than in the Non-Tg/lean. In the DIO groups, the levels were lower in the Tg^MEK6^/DIO than in the Non-Tg/DIO. For TNFα, a representative pro-inflammatory cytokine, Tg^MEK6^/lean exhibited a significant increase compared to the Non-Tg/lean. IL-10, with anti-inflammatory functions, showed a decrease (Figure 7C,D). Thus, MEK6 overexpression was shown to promote the differentiation of M1 to increase the production of pro-inflammatory cytokines and inhibit the anti-inflammatory cytokines of M2.

## 3. Discussion

Based on the hypothesis that *MEK6* overexpression impairs the control of energy consumption to induce obesity, we produced a novel *MEK6* overexpression model and investigated the effects of the interaction between HFD and *MEK6*. We found the interacting effects of *MEK6* with HFD on fat accumulation and inflammatory cytokines production in the liver, WAT, and BMDM. The MEK6 expression reduced lipolysis in WAT, thermogenesis in the liver, and WAT and M2-associated cytokines in BMDM (Figure 8). Plasma ALT and AST levels or AST/ALT ratio in all groups were fell within the normal range. Our experiments were done with the lack of hepatotoxicity, although AST and ALT were slightly increased by *MEK6* expression and HFD [34,35]. 

*MEK6* overexpression was shown to induce lipid accumulation and hypertrophy in adipocytes. WATs are elaborate organs with a significant role in energy homeostasis, and the quality of the adipose tissue is mainly determined by adipocyte number and size [36]. The number of adipocytes is adjusted and finalized in childhood, and the increase in adipocyte number in adulthood may be a key mechanism in energy imbalance [37]. *MEK6* overexpression increased cell size in the adipose tissue with hypertrophy to a greater degree in the *MEK6* overexpression rather than non-Tg in both lean and DIO groups. The expression of fat synthesis factors (PPAR-γ, C/EBP-α, SREBP1) and the histological examination of lipid accumulation supported the effect of the *MEK6* in promoting adipogenesis [38,39]. The adipocyte hypertrophy with *MEK6* overexpression appeared to contribute to the energy imbalance by increasing adipocyte size via excess TG accumulation in WATs. The expression of PPARγ and SREBP1 as the key proteins in adipogenesis was upregulated to increase lipogenesis in the liver.

In mice with HFD-induced obesity livers, *MEK6* reduced fatty acid oxidation and exacerbated lipid accumulation. This is consistent with the results of increased levels of ERK and MEK in ERK KO mice [40], and the ERK activity in the liver increased MEK expression for the downregulation of p38 phosphorylation [41]. Both PPARα and UCP2 could exhibit different expressions according to the level of obesity, and in the state of high lipid accumulation because of HFD, PPARα and UCP2 expression could have increased as an adaptive response to prevent the generation of ROS in the mitochondria [42,43]. Therefore, the promotion of PPARα expression with *MEK6* expression was an adaptive response to excess lipid metabolism. At the same time, UCP2 participated in ROS generation and immune cell activation via the MAPK pathway in macrophages [44,45,46], which showed a decline in its expression because of reduced p38 activity with *MEK6* expression. Thus, *MEK6* overexpression mediated a decline in p38 activity in the liver to suppress energy consumption because of reduced fatty acid oxidation, whereas it also induced lipid accumulation. The response may be further exacerbated through the interaction with the HFD.

*MEK6* induces inhibition of UCP1, which is closely related to thermogenesis by inhibiting the activity of p38. This has been shown to exacerbate obesity by inhibiting the thermogenesis of adipose tissue. The fall in ERK/p38 phosphorylation caused by the *MEK6* expression in WATs was correlated with the regulation of adipogenesis and thermogenesis [47,48,49]. HSL releases free fatty acids from the stored lipids in adipocytes to initiate the fatty acid oxidation pathway and lipolysis [50]. UCP1, in response to the continuous stimulation of lipolysis, promoted thermogenesis in the mitochondria. This is harnessed in WATs to control the balance in the bidirectional interconversion of brite/beige adipose tissue (BAT) [51,52]. In this study, *MEK6* expression led to a reduced ERK/p38 phosphorylation in eWATs to downregulate HSL and UCP1 expression. In previous studies, the *MEK6* KO mice showed an increase in p38 activity, and the *ERK* KO mice showed increased MEK activity [23,40]. This coincided with the results observed in 3T3-L1-induced beiging adipocytes, where the levels of adipokines and inflammatory cytokines decreased because of *MEK6* transfection [24]. The low level of p38 activity was shown to control UCP1 and have a regulatory role in thermogenesis [48,49]. Hence, the fall in p38 activity because of *MEK6* overexpression led to low HSL and UCP1, reducing lipolysis and thermogenesis. This effect was amplified through the synergistic interaction with HFD.

*MEK6* overexpression has a role in regulating cytokines in adipose tissues and macrophages. In inflammatory adipose tissues, the number of macrophages is proportional to the level of inflammation [53], and the accumulation of BMDM was shown to be the main cause of chronic and systemic inflammation in obesity [54]. The decline in the adiponectin/leptin ratio (adipokines) in the adipose tissue and plasma with *MEK6* expression led to a further decrease in MEK6 overexpressed groups with HFD. The IL-10/TNFα ratio exhibited a similar trend. Leptin is a protein secreted by adipocytes to control energy metabolism from suppressing appetite and promoting energy consumption in the body [55]. A positive correlation was reported between leptin secretion and the increased size of adipocytes and lipid contents caused by resistance [56,57]. However, a negative correlation occurred between fat accumulation and the decline in the plasma level of adiponectin in both obese males and females [58], in agreement with the increase in adipogenesis factors with *MEK6* overexpression. This indicated the role of *MEK6* in the regulation of lipid accumulation and adipokine expression. TNFα, in particular, because it is secreted in obese individuals, could be a cause of suppressed synthesis or secretion of adiponectin [59], which indicates a close correlation between cytokines and changes in adipokines because of lipid accumulation in the body. In this study, the expression of pro-inflammatory cytokines increased in M1, and anti-inflammatory cytokines decreased in M2 for the BMDM in the *MEK6* overexpression rather than in non-Tg in both lean and DIO groups. Adiponectin, as a regulatory factor in systemic homeostasis, promotes the differentiation of M2 macrophages and hence mediates their activities [60,61]. Adiponectin has also been shown to promote the proliferation of M2 macrophages to improve browning, which is related to the decline in UCP1 with *MEK6* expression [62]. Thus, the decline in the adiponectin/leptin ratio because of *MEK6* overexpression leads to the promotion of M1 and suppression of M2 [61,63,64], which indicates that *MEK6* contributes to the regulation of inflammatory or anti-inflammatory factors in the adipose tissue and BMDM and that such negative effects might be further amplified through the synergistic interaction with the HFD.

The limitation in this study was the lack of complementation through the analysis of BAT tissues, based on the result of UCP2, the liver lipolysis marker, and UCP1, the thermogenesis marker in the adipose tissue. However, it is significant that an in vivo study was conducted based on the findings of previous studies on *MEK6* that used human and 3T3-L1 cells [15,24] and reported a regulatory role of *MEK6* in p38 activity and M1/M2 inflammatory factors to suppress lipolysis and thermogenesis but induce lipid accumulation and inflammation thereby exacerbating the state of obesity. RNA expression of the *MEK6* gene was expressed in various organs such as the liver, heart, and fat as same results in this PCR study. Selected liver and WAT as target organs to confirm the interaction between *MEK6* and HFD, we confirmed the lipid metabolism in those organs. In further research studies, we will design the inter-mechanisms study in *MEK6* over-expressed organs for potential therapeutics. In summary, the results in this study provided evidence for the interaction between *MEK6* and an HFD, which leads to physiological changes in the accumulation of obesity biomarkers. The data are anticipated to be useful in the development or monitoring of early diagnostic markers for obesity.

## 4. Materials and Methods

### 4.1. Animal Design

In this study, male C57BL/6N mice (Non-Tg, DBL, Chungbuk, Korea) and transgenic mice (Tg, Macrogen, Seoul, Korea) were used. All mice were bred using the 12 h light-dark cycle under the following conditions: 21 ± 1 °C temperature and 55 ± 1 °C humidity. The mice were induced into two groups: Lean and DIO (diet-induced obesity) using a Chow diet (CD, DBL Co., Cat #RodFeed, Chungbuk, Korea) with 3.5% fat content and a HFD (Cat #D12492, Research Diets, Inc., New Brunswick, NJ, USA) with 60% fat content. The experimental design comprised four groups: Non-Tg/lean; Non-Tg/DIO; Tg*^MEK6^*/lean; Tg*^MEK6^*/DIO. During the 15-week breeding period, the body weight, dietary intake, and food efficiency ratio (FER) were recorded each week. The FER was calculated as body weight gain (g)/food intake (g). The animal experiments were conducted following the review and approval of the Sungshin Women’s University IACUC (No. SSWIACUC-2019-001-01).

### 4.2. MEK6 Overexpression and PCR Genotyping in C57BL/6N Mice

The *MEK6* overexpression transgenic mice (Tg*^MEK6^*) were produced using the *MEK6* vector and pCMV6-AC-GFP plasmid (clone, Cat #MG204942, Origene, Rockville, MD, USA) in C57BL/6N background mice. The transgene *MEK6* was expressed by CMV promoter. The production of Tg*^MEK6^* was performed in collaboration with the Humanizing Genomics Macrogen (Macrogen Inc., Seoul, Korea). The purified DNA (4 ng/μL) was microinjected into the fertilized embryo in a superovulated female, and in vitro fertilization (IVF) was performed with the cultured F0 male mice sperm and female oocytes to produce F1. The insertion of the genomic DNA in Tg*^MEK6^* mice was confirmed through PCR, using an excised section of the tail, prior to their use in subsequent experiments.

To confirm the *MEK6* mRNA expression in the tissues of the Tg*^MEK6^* model, the TRizol reagent (Cat #15596026, ThermoFisher, Waltham, MA, USA) was used. After extracting a 0.1 g sample each from the liver, heart, epididymal fat (eWAT), perineal (pWAT), and subcutaneous fat (sWAT), tissues were homogenized, and the RNA pellets were isolated by chloroform (Cat #C2432-1L, Sigma-Aldrich, St. Louis, MO, USA) and isopropanol (Cat #D858, Duksan, Seoul, Korea). The resulting RNA pellets were dissolved in RNase-free water and heated at 65℃ for 10 min. The RNA was reverse transcribed into cDNA using the ReversTra Ace PCR kit (Cat #FSQ-101, TOYOBO, Osaka, Japan). *MEK6* vector-specific primer was produced for amplification using the reverse transcription-polymerase chain reaction (RT-PCR) (Primer; Mitogen-activated protein kinase kinase 6 (MEK6, forward: 5′-GTGGATAGCGGTTTGACTCAC-3′), reverse: 5′-GTAGAAGGTCACGGTGAATGG-3′), glyceraldehyde-3-phosphate dehydrogenase (GAPDH, forward: 5′-CGTGCCGCCTGGAGAAACC-3′, reverse: 5′-TGGAAGAGTGGGAGTTGCTGTTG-3′). The images from the Chemidoc Imaging System (Bio-Rad, Hercules, CA, USA) were analyzed and quantified using Image Lab Software (Bio-Rad, Hercules, CA, USA).

### 4.3. Histological Analysis of the Liver and Adipose Tissue

Mice were sacrificed, and the liver and sWAT were extracted, from which blood was removed before fixation using 4% formaldehyde (Cat #d717, Duksan, Seoul, Korea). The fixed tissues were pretreated for the paraffin formatting to generate 5–6 μm thick slices. Hematoxylin (Cat #03971, Sigma-Aldrich, St. Louis, MO, USA) and eosin (Cat #3600, BBC, Mount Vernon, WA, USA) (H&E) were used to obtain stained sections, and the cross-section was observed under an optical microscope (Cat #BX60, Olympus, Tokyo, Japan). Adiposoft of the ImageJ software (version 1.53e, Java 1.8.0_172, Wayen Rasband, U.S. NIH, Bethesda, MD, USA) was used to quantify the cross-section of each tissue.

### 4.4. Bone-Marrow-Derived Macrophages (BMDM) Isolation and Differentiation

A thighbone was removed to extract the primary bone marrow cells from 22-week-old C57BL/6N mice. The cells were cultured in a 37 °C, in a 5% CO_2_ incubator, using the RPMI 1640 Dulbecco’s modified Eagle’s medium (DMEM, Cat #LM011-03, Welgene, Deagu, Korea) containing 30% L929 conditioned medium with macrophage-colony stimulatory factor (M-CSF), 20% fetal bovine serum (FBS, Capricorn Scientific GmbH, Ebsdorfergrund, Germany), and 1% Penicillin/streptomycin (P/S). For the differentiated macrophages (BMDM, unstimulated, M0) to form a monolayer, the medium was replaced once in 3 d for 7 d. The adhering BMDMs were treated with 50 ng/mL of lipopolysaccharide (LPS Cat #L3024, Sigma-Aldrich, St. Louis, MO, USA) and 20 ng/mL of interleukin 4 (IL-4 Cat #SRP3211, Sigma-Aldrich, St. Louis, MO, USA) for 24 h, to induce M1 and M2 macrophages, which were subsequently harvested using the Gibco^®^ Versene solution (Cat # 15040066, ThermoFisher, Waltham, MA, USA). The expression of the macrophage markers (COX-2, CD86, Arginase-1, TNFα, C137L, IL-6, IL-10) was measured through western blotting or ELISA.

### 4.5. Biochemical Parameters 

Blood glucose was measured immediately after blood collection, using a blood glucose meter (SD Codefree, Cat #01GM11, SD Co., Suwon, Korea). Plasma was isolated from whole blood using a centrifuge (13,000 rpm, 20 min). The measurements of aspartate transaminase (AST, Cat #AM103-K), alanine aminotransferase (ALT, Cat #AM102), triglycerides (TGs, Cat #AM157S), and total cholesterol (TC, Cat #AM202) in the plasma and liver tissue were performed using the respective quantification kits (ASAN Pharm Corporation, Seoul, Korea). The reference concentration was measured to draw the standard curve, and the sample absorbance was measured using a microplate reader at 505 nm, 550 nm, and 500 nm. For the liver, the tissue was pulverized, and the Folch method [65] was applied using chloroform (Cat #C2432-1L, Sigma-Aldrich, St. Louis, MO, USA) and methanol (Cat #, Merck, Darmstadt, Germany) to obtain a lipid solution. The solution was used after treatment with nitrogen gas.

The concentrations of IL-10 and TNF-α in the plasma or culture supernatant were both measured using ELISA kits (Cat #431414-430904, BioLegend, San Diego, CA, USA) through the standard protocol. The unplates were 96-well plate (Cat #423501) coated overnight with a coating buffer (Cat #79008) containing IL-10 or TNF-α, and the 1X assay diluent (Cat #78888) was used for 2 h for blocking. After 1 h adherence through cell culture supernatant and plasma, the secondary antibodies were incubated. After the adherence of Avidin-HRP (Cat #79004), the tests were performed using the substrate solution (Cat #78570-78571).

### 4.6. Preparation of Protein Extracts and Western Blot Analysis

The homogenization of the frozen tissues and cells using the grinder was followed by dissolution in cold radioimmune precipitation (RIPA) buffer (Cat #EBA-1149, Elpis Biotech, Daejeon, Korea) containing the protease inhibitor and phosphatase inhibitor. The protein sample was prepared and quantified using the Bradford method [65] and heated at 100 ℃ for 10 min. The proteins were extracted through 8–12% sodium dodecyl sulfate-polyacrylamide gel (SDS-PAGE) electrophoresis. After isolation, they were transferred to a polyvinylidene fluoride membrane (PVDF, GE Healthcare, Chicago, IL, USA). The membrane was treated with 3% bovine serum albumin (BSA, Cat #BSA100, Bovogen, Melbourne, Australia) and blocked for 1 h, followed by overnight incubation with primary antibodies diluted to 1:1000 at 4 °C, including β-actin (Cat #E12-041, Enogene, Jiangsu, China), vinculin (Cat #4650S, CST, Beverly, MA, USA), pMEK6 (Cat #9236S, CST, Beverly, MA, USA), MEK6 (Cat #MA5-15808, Invitrogen, Carlsbad, CA, USA), pERK (Cat #9101S, CST, Beverly, MA, USA), ERK (Cat #9102S, CST, MA, USA), pp38 (Cat #4511T, CST, Beverly, MA, USA), p38 (Cat #8690T, CST, Beverly, MA, USA), pACC (Cat #3661S, CST, Beverly, MA, USA), ACC (Cat3676S #, CST, Beverly, MA, USA), pHSL (Cat #4139S, CST, Beverly, MA, USA), HSL (Cat #4107S, CST, Beverly, MA, USA), C/EBPα (Cat #SC-9314, Santa cruz, Dallas, TX, USA), PPARγ (Cat #SC-7273, Santa cruz, Dallas, TX, USA), SREBP1 (Cat #, Santa cruz, Dallas, TX, USA), adiponectin (Cat #2789S, CST, Beverly, MA, USA), leptin (Cat #675002, Biolegend, San Diego, CA, USA), PPARα (Cat #sc-398394, Santa cruz, Dallas, TX, USA), UCP1 (Cat #14670S, CST, Beverly, MA, USA), UCP2 (Cat #89326S, CST, Beverly, MA, USA), COX-2 (Cat #12282S, CST, Beverly, MA, USA), CD86 (Cat #NB110-55488, Novus biological, Littleton, CO, USA), IL-6 (Cat #5145-100, Biovision, Palo Alto, CA, USA), arginase1 (Cat #SC-20150, Santa cruz, Dallas, TX, USA), and CD137L (Cat #SC-11819, Santa cruz, Dallas, TX, USA). Next, the membrane was incubated with secondary antibodies diluted to 1:5000 for 1 h. The Chemidoc Imaging System (Bio-Rad, Hercules, CA, USA) was used for chemiluminescence with the ECL solution, and the result was analyzed using the Image Lab Software (Bio-Rad, Hercules, CA, USA).

### 4.7. Statistical Analysis

All data were statistically analyzed using the Statistical Package for the Social Science software ver. 25.0 (SPSS Inc., Chicago, IL, USA). The significance of the between-group difference was compared using Student’s t-test or Duncan’s multiple range test following an ANOVA. Data were expressed as the mean ± standard deviation (SD). The level of significance for mean variation was set to *p* < 0.05. The graph was produced using GraphPad Prism (version 8.0.1, GraphPad Software, La Jolla, CA, USA), where the letters a–c, and d indicate significant differences for each of the four experimental groups in the comparison.

## Figures and Tables

**Figure 1 ijms-22-13559-f001:**
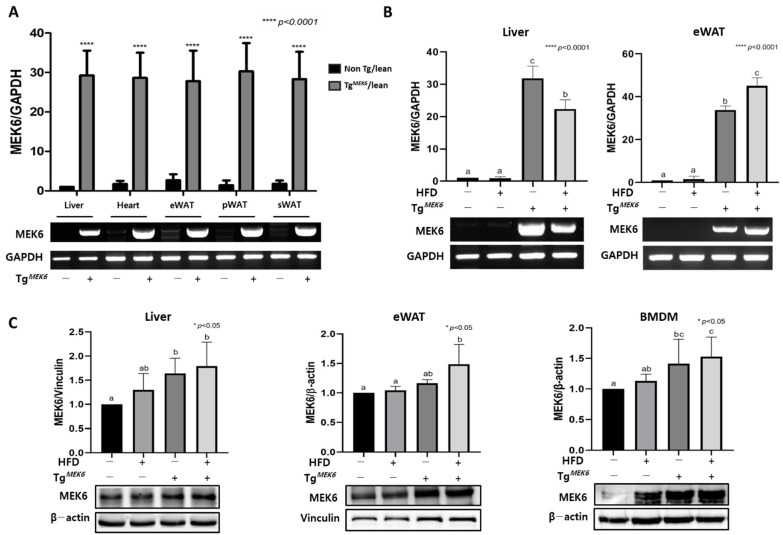
Generation of a mouse model overexpressing *MEK6*. (**A**) Relative expression of *MEK6* mRNA in the indicated tissues of Tg*^MEK6^*/lean (*n* = 10) compared to non-Tg/lean (*n* = 10); Liver, Heart, eWAT (Epididymal fat), pWAT (Perirenal fat), sWAT (Subcutaneous fat). (**B**) Relative expression of *MEK6* mRNA in Liver and eWAT of non-Tg/lean (*n* = 10), non-Tg/DIO (*n* = 10), Tg*^MEK6^*/lean (*n* = 10) and Tg*^MEK6^*/DIO (*n* = 10). (**C**) Expression levels of MEK6 proteins in liver, eWAT and BMDM. Results are expressed as mean ± SD. Data were assessed by Student’s t-test or one-way ANOVA with Duncan’s test (NS, not significant by unpaired *t*-test. *; *p* < 0.05, ****; *p* < 0.0001). Superscript letters (a–c) indicate the significant differences between the groups at *p* < 0.05. +, treatment or transfection; −, non-treatment or non-transfection.

**Figure 2 ijms-22-13559-f002:**
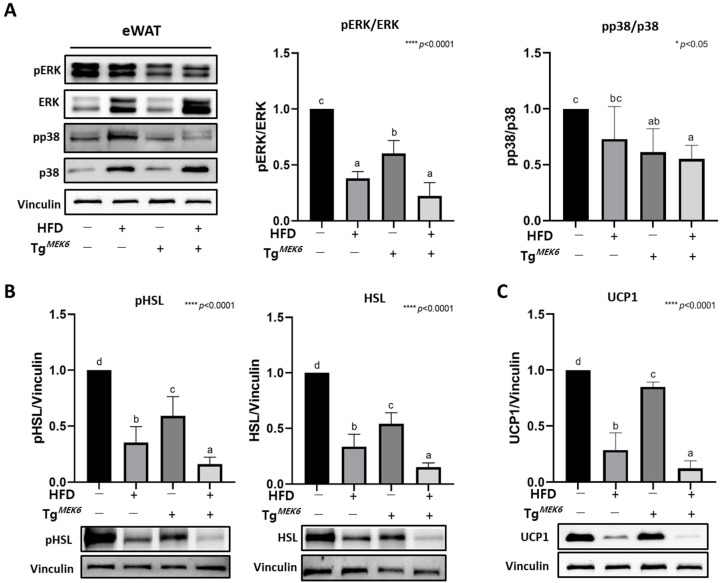
Overexpression of *MEK6* modulates MAPK family (ERK/p38) activation and protein expression related to lipolysis and thermogenesis in eWAT. (**A**) Expression of proteins (pERK, ERK, pp38, p38) related to MAPK family in eWAT. (**B**) Expression of proteins (pHSL, HSL) related to lipolysis and (**C**) protein (UCP1) related to beiging in eWAT. Results are expressed as mean ± SD. Data were assessed by one-way ANOVA with Duncan’s test (NS, not significant by unpaired *t*-test. *; *p* < 0.05, ****; *p* < 0.0001). Superscript letters (a–d) indicate the significant differences between the groups at *p* < 0.05. +, treatment or transfection; −, non-treatment or non-transfection.

**Figure 3 ijms-22-13559-f003:**
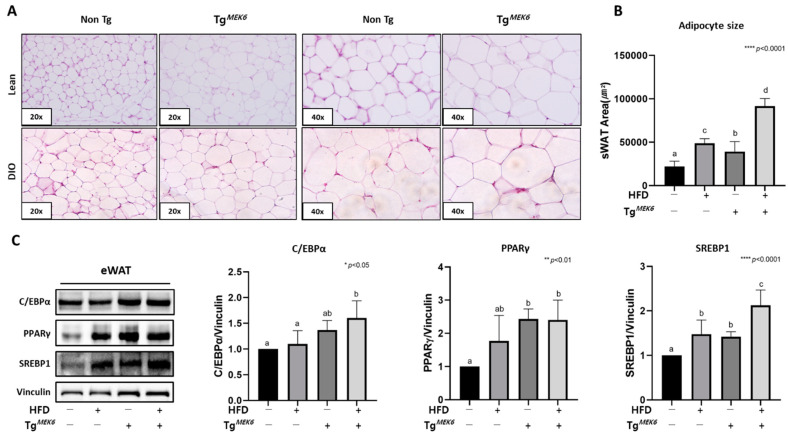
*MEK6* overexpression of WAT was increased adipocyte hypertrophy and adipogenesis in C57BL/6 mice. (**A**) Representative sections of H&E staining show the adipocytes in sWAT (subcutaneous fat); 20×, 40× magnification (**B**) Quantification of adipocyte size in sWAT. (**C**) The expression of proteins related to adipogenesis (C/EBPα, PPARγ) and cholesterol synthesis (SREBP1) in eWAT (epididymal fat). Results are expressed as mean ± SD. Data were assessed by one-way ANOVA with Duncan’s test (NS, not significant by unpaired *t*-test. *; *p* < 0.05, **; *p* < 0.01, ****; *p* < 0.0001). Superscript letters (a, b, c, d) indicate the significant differences between the groups at *p* < 0.05. +, treatment or transfection; −, non-treatment or non-transfection.

**Figure 4 ijms-22-13559-f004:**
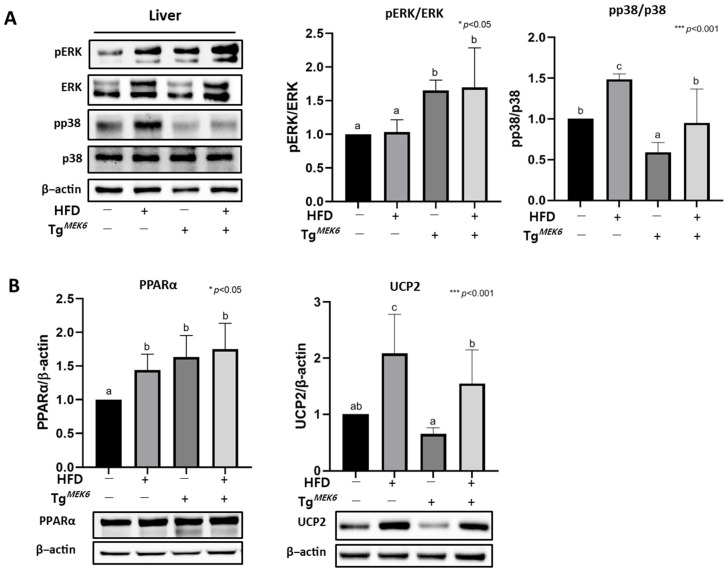
Overexpression of *MEK6* modulates protein related to MAPK family (ERK/p38) activation and PPARα, UCP2 in liver. (**A**) The liver of Tg*^MEK6^* mice was confirmed makers (pERK, ERK, pp38, p38) related to MAPK family and (**B**) protein expression of PPARα, UCP2 by western blot. Results are expressed as mean ± SD. Data were assessed by one-way ANOVA with Duncan’s test (NS, not significant by unpaired *t*-test. *; *p* < 0.05, ***; *p* < 0.001). Superscript letters (a–c) indicate the significant differences between the groups at *p* < 0.05. +, treatment or transfection; −, non-treatment or non-transfection.

**Figure 5 ijms-22-13559-f005:**
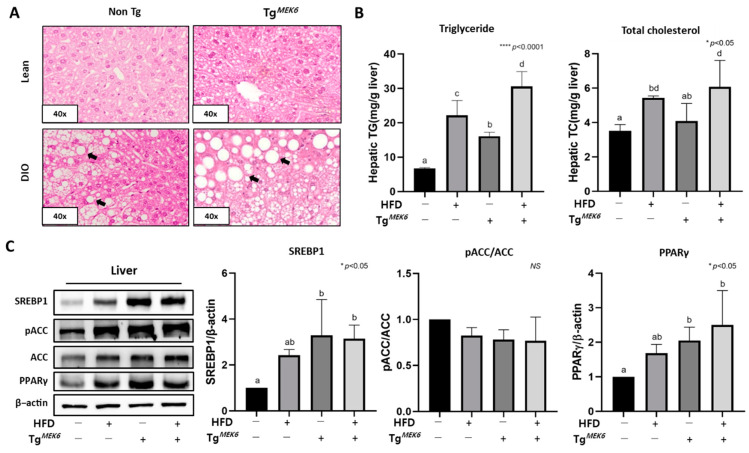
*MEK6* overexpression promotes lipid accumulation in liver. (**A**) Representative sections of H&E staining show lipid accumulation and lipid droplet (black arrows) in liver; 40x magnification (**B**) Triglyceride (TG) and total cholesterol (TC) levels was measured in the lipid-extracts of liver. (**C**) The expression of proteins (SREBP1, pACC/ACC and PPARγ) related to lipid accumulation were confirmed in the liver by western blotting. Results are expressed as mean ± SD. Data were assessed by one-way ANOVA with Duncan’s test (NS, not significant by unpaired *t*-test. *; *p* < 0.05, ****; *p* < 0.0001). Superscript letters (a–d) indicate the significant differences between the groups at *p* < 0.05. +, treatment or transfection; −, non-treatment or non-transfection.

**Figure 6 ijms-22-13559-f006:**
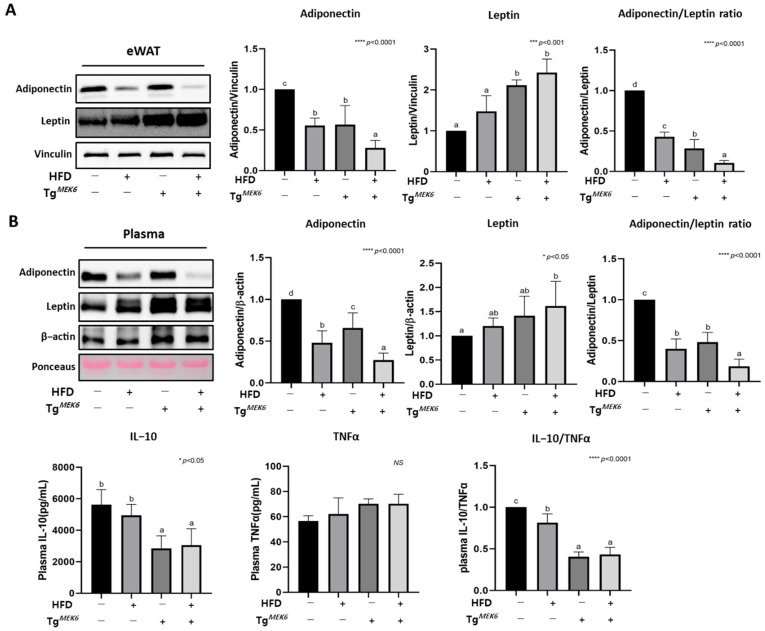
Effects of *MEK6* overexpression on the cytokines of WAT and plasma. (**A**) The expression of proteins related to adipokines (adiponectin, leptin) and adiponectin/leptin ratio were measured by western blotting in eWAT. (**B**) The expression of proteins related to adipokines (adiponectin, leptin) and adiponectin/leptin ratio and cytokine levels such as IL-10(anti-inflammatory), TNFα(pro-inflammatory) and IL-10/TNFα were measured by western blotting or ELISA in plasma. Results are expressed as mean ± SD. Data were assessed by one-way ANOVA with Duncan’s test (NS, not significant by unpaired *t*-test. *; *p* < 0.05, ***; *p* < 0.001, ****; *p* < 0.0001). Superscript letters (a–d) indicate the significant differences between the groups at *p* < 0.05. +, treatment or transfection; −, non-treatment or non-transfection.

**Figure 7 ijms-22-13559-f007:**
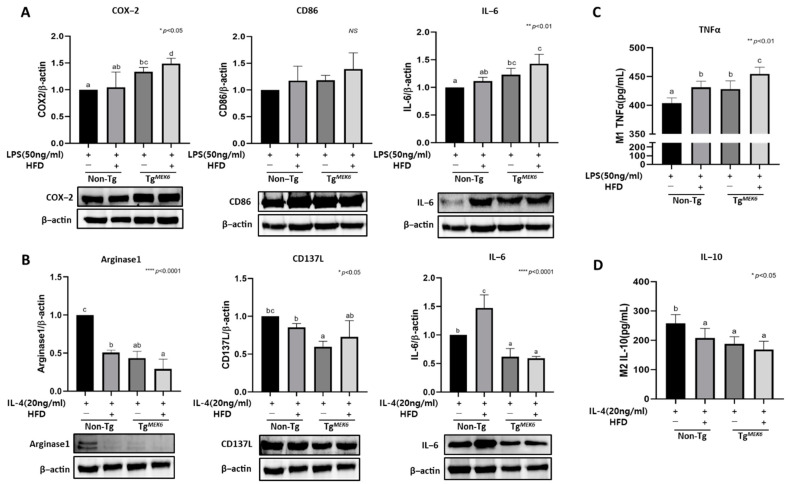
Effects of *MEK6* on inflammatory cytokines of macrophage in BMDM. (**A**) The LPS (50 ng/mL)-induced M1 macrophage confirmed expression of proteins related to pro-inflammatory (COX-2, CD86, IL-6) by western blot. (**B**) The IL-4 (20 ng/mL)-induced M2 macrophage confirmed expression of proteins related to anti-inflammatory (Arginase-1, CD137L, IL-6) by western blot. (**C**,**D**) Cytokine levels of M1 and M2 culture supernatant such as IL-10 (anti-inflammatory) and TNF-α(pro-inflammatory) were measured by ELISA. Results are expressed as mean ± SD. Data were assessed by one-way ANOVA with Duncan’s test (NS, not significant by unpaired *t*-test. *; *p* < 0.05, **; *p* < 0.01, ****; *p* < 0.0001). Superscript letters (a–d) indicate the significant differences between the groups at *p* < 0.05. +, treatment or transfection; −, non-treatment or non-transfection.

**Figure 8 ijms-22-13559-f008:**
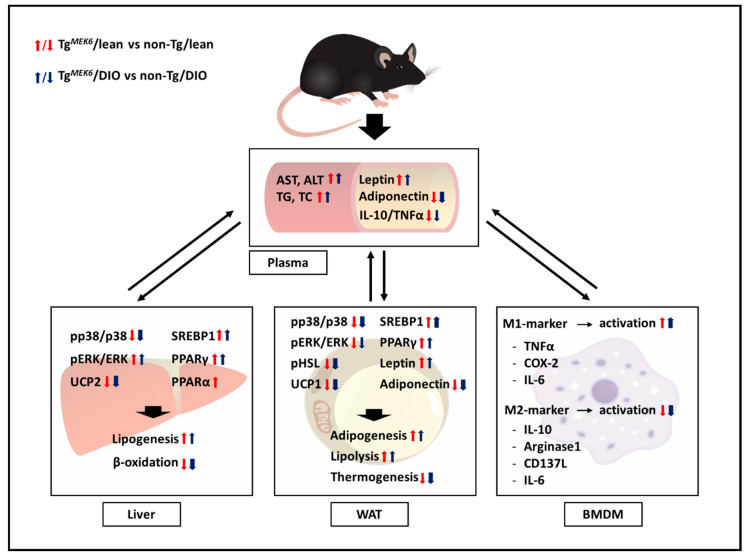
Regulation of lipid accumulation and Inflammation in Tg*^MEK6^*/lean and Tg*^MEK6^*/DIO. Red allows show the effect of Tg*^MEK6^*/lean compared to non-Tg/lean, and blue allows show an effect of Tg^*MEK6*^/DIO compared to non-Tg/DIO. Thick allows indicate cases with large differences. Case where there was no change was not expressed.

**Table 1 ijms-22-13559-t001:** Body weight, total body weight, total food intakes, food efficiency and tissue weight.

Group		Non_Tg		Tg*^MEK6^*		*p*-Value
		Lean (n = 10)	DIO (n = 10)	Lean (n = 10)	DIO (n = 10)	
**Body Weight (g)**	**Initial**	19.82 ± 0.82 ^a^	22.34 ± 0.73 ^b^	23.12 ± 1.23 ^b^	23.26 ± 1.00 ^b^	<0.0001 ****
	**Final**	30.75 ± 1.54 ^a^	49.72 ± 1.76 ^c^	33.56 ± 2.27 ^b^	52.44 ± 1.72 ^d^	<0.0001 ****
**Body weight gain (g/day)**		0.11 ± 0.02 ^a^	0.28 ± 0.02 ^b^	0.11 ± 0.02 ^a^	0.30 ± 0.09 ^c^	<0.0001 ****
**Total body weight gain (g)**		10.92 ± 1.83 ^a^	27.39 ± 1.29 ^b^	10.44 ± 1.47 ^a^	29.18 ± 1.92 ^c^	<0.0001 ****
**Dietary intake (g/day body weight)**		0.10 ± 0.01 ^b^	0.05 ± 0.01 ^a^	0.11 ± 0.01 ^c^	0.05 ± 0.01 ^a^	<0.0001 ****
**Total dietary intake (g/body weight)**		10.52 ± 0.65 ^b^	5.25 ± 0.29 ^a^	11.88 ± 0.70 ^c^	5.72 ± 0.30 ^a^	<0.0001 ****
**Food efficiency ratio**		1.05 ± 0.23 ^a^	5.23 ± 0.44 ^b^	0.89 ± 0.16 ^a^	5.12 ± 0.55 ^b^	<0.0001 ****
**Tissue weight (g/100g bodyweight)**						
**Liver**		3.74 ± 0.50 ^a^	4.45 ± 0.69 ^bc^	3.99 ± 0.17 ^ab^	4.67 ± 1.00 ^c^	<0.05 *
**Heart**		0.36 ± 0.04 ^b^	0.25 ± 0.02 ^a^	0.43 ± 0.04 ^c^	0.26 ± 0.03 ^a^	<0.0001 ****
**Kidney**		1.19 ± 0.14 ^c^	0.77 ± 0.07 ^a^	1.27 ± 0.09 ^c^	0.87 ± 0.10 ^b^	<0.0001 ****
**Total fat**		2.92 ± 1.11 ^a^	7.77 ± 0.99 ^b^	3.42 ± 1.57 ^a^	7.50 ± 0.91 ^b^	<0.0001 ****
**Perirenal fat (pWAT)**		0.48 ± 0.27 ^a^	2.25 ± 0.58 ^b^	0.88 ± 0.36 ^a^	2.67 ± 0.39 ^b^	<0.0001 ****
**Visceral fat (vWAT)**		0.85 ± 0.46 ^a^	3.05 ± 0.61 ^d^	0.66 ± 0.30 ^a^	2.43 ± 0.59 ^c^	<0.05 *
**Epididymal fat (eWAT)**		1.59 ± 0.73 ^a^	2.46 ± 0.32 ^b^	1.88 ± 1.05 ^ab^	2.41 ± 0.76 ^b^	<0.0001 ****

Body weight, total body weight, total food intakes, food efficiency and tissue weight in 22-week-old non-Tg and Tg*^MEK6^*. Results are expressed as mean ± SD. All data were assessed by one-way ANOVA with Duncan’s test (NS, not significant by unpaired *t*-test. *; *p* < 0.05, ****; *p* < 0.0001). Superscript letters (a–d) indicate the significant differences between the groups at *p* < 0.05.

**Table 2 ijms-22-13559-t002:** Blood chemistry parameters in non-Tg and Tg*^MEK6^*.

Group	Non Tg		Tg*^MEK6^*		*p*-Value
	Lean (n = 10)	DIO (n = 10)	Lean (n = 10)	DIO (n = 10)	
**Blood glucose (mg/dL)**	119.1 ± 25.02 ^a^	185.71 ± 36.09 ^b^	123.22 ± 29.75 ^a^	199.94 ± 48.57 ^b^	<0.0001 ****
**AST (IU/L)**	47.48 ± 1.62 ^a^	63.83 ± 1.14 ^b^	59.29 ± 3.76 ^c^	76.46 ± 1.64 ^d^	<0.001 ***
**ALT (IU/L)**	8.62 ± 0.40 ^a^	48.95 ± 1.78 ^b^	15.38 ± 0.70 ^c^	71.74 ± 5.30 ^d^	<0.001 ***
**AST/ALT ratio**	5.51 ± 0.25 ^c^	1.30 ± 0.04 ^a^	3.86 ± 0.15 ^b^	1.07 ± 0.06 ^a^	<0.001 ***
**TG (mg/dL)**	60.50 ± 2.74 ^a^	77.61 ± 3.46 ^c^	70.25 ± 4.13 ^b^	89.47 ± 3.86 ^d^	<0.01 **
**TC (mg/dL)**	146.08 ± 3.71 ^a^	243.92 ± 9.32 ^c^	187.92 ± 7.13 ^b^	285.00 ± 2.83 ^d^	<0.0001 ****

Blood biochemistry in 22-week-old non-Tg and Tg*^MEK6^*. Results are expressed as mean ± SD. All data were assessed by one-way ANOVA with Duncan’s test (NS, not significant by unpaired *t*-test. **; *p* < 0.01, ***; *p* < 0.001, ****; *p* < 0.0001). Superscript letters (a–d) indicate the significant differences between the groups at *p* < 0.05.

## Data Availability

Not applicable.

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
