# Peer review of "MEK6 Overexpression Exacerbates Fat Accumulation and Inflammatory Cytokines in High-Fat Diet-Induced Obesity"

_ijms, 2021, doi:10.3390/ijms222413559_

Round 1

Reviewer 1 Report

In their manuscript “MEK6 overexpression synergistically exacerbates inflammatory cytokines and fat accumulation in high-fat diet-induced obesity” Lee et. al. evaluate the metabolic phenotype of overexpression of MEK6, an important MAPKK that plays a role in the cellular response to various stresses. The authors show, using a Tg-MEK6 mouse model, that MEK6 activity exacerbates the metabolic effect induced by HFD feeding. This includes increased body weight, increased levels of serum TG and cholesterol, adipose tissue hypertrophy, and fat accumulation in the liver.

While the metabolic phenotype is robust, the biological importance is questionable since overexpression of such an important kinase is likely to result in phosphorylation of non-specific sites, which might not be physiologically relevant. Moreover, despite being an over-expression model, the upregulation of MEK6 protein, at least in the liver, seems very minor and it is not clear to what extent it actually contributes to the observed phenotype.

Major concerns –

  1. The authors suggest a synergistic effect of MEK6 and HFD. However, in most cases, their data indicates an additive effect rather than a synergistic effect. This should be corrected.
  2. It is not clear under which promoter the transgene MEK6 is being expressed. This should be clarified in the methods section.
  3. In Fig. 1B, please show the overexpression using qPCR, similar to Fig. 1A.
  4. Under what conditions the plasma biochemistry analysis was done, is this fasting?
  5. The authors should provide an explanation of why the p38 phosphorylation is reduced even though its kinase is being over-expressed.

Author Response

We appreciate the Reviewer’s comments. The followings are our point-by-point responses.

Major concerns 

1. The authors suggest a synergistic effect of MEK6 and HFD. However, in most cases, their data indicates an additive effect rather than a synergistic effect. This should be corrected.
-> ANS: Since we agree with the reviewer’s comment and “synergistic effect” was not propriate in some of the results, we excluded it the title and main text. 

2. It is not clear under which promoter the transgene MEK6 is being expressed. This should be clarified in the methods section.
-> ANS: The MEK6 vector used for MEK6 overexpression was pCMV6-AC-GFP plasmid, and the CMV promoter expressed the transgene MEK6. This content was added to the method, line 384. 

3. In Fig. 1B, please show the overexpression using qPCR, similar to Fig. 1A.
-> ANS: We added the PCR graph in Figure 1B.

4. Under what conditions the plasma biochemistry analysis was done, is this fasting?
-> ANS: Blood was collected after fasting for 18 hrs and the biochemistry was measured in plasma.

5. The authors should provide an explanation of why the p38 phosphorylation is reduced even though its kinase is being over-expressed.
-> ANS: There are two major upstream kinases of p38, MEK6 and MEK3, and MEK6 can activate the most of p38 family. p38α is preferentially activated by low-MEK6 concentrations, and an appropriate ratio of MEK6 to p38α is known to be involved in the specificity of signaling by p38α. According to reports that MEK6 was upregulated in p38 KO mice, there is negative correlation between MEK6 and p38α expression/activity on the cellular responses such as inflammatory signaling. See the following reports.

1)Ambrosino, Concetta, et al. "Negative feedback regulation of MKK6 mRNA stability by p38α mitogen-activated protein kinase." Molecular and cellular biology 23.1 (2003): 370-381.
2) Inoue, Tomoyuki, et al. "Mitogen-activated protein kinase kinase 3 is a pivotal pathway regulating p38 activation in inflammatory arthritis." Proceedings of the National Academy of Sciences 103.14 (2006): 5484-5489. 

Thank you so much for insightful comments.

Reviewer 2 Report

Lee et al. present a well-performed, clearly-written manuscript where they investigated the role of MEK6 in the context of obesity and its associated mechanisms (lipid accumulation and inflammation). Personally, I believe the research is of a high level. In case the authors can clarify some questions (indicated below), I fully support publication.

Major comments:

  • Besides the described effects on lipid metabolism and inflammation, I can imagine that manipulation of MEK6 also has influence on other metabolic aspects (for instance maybe insulin signaling?). To what extent do the authors believe that the metabolic effects of MEK6 are related to lipids and do they believe that also other metabolic aspects are involved? (some elaboration on this matter in the discussion of the manuscript would be interesting)

  • The model the authors use is a ‘full-body’ overexpression of MEK6, making it unclear what the involved of MEK6 is in the respective metabolic organs (liver, heart, WAT). Could the authors elaborate on this matter in the discussion as well? Especially in the context of potential future therapeutics, I think this could have an added value to the paper.

  • Line 86: ‘In contrast, MEK6 protein expression in the liver and eWATs was higher in the Non-Tg/DIO than in Non-Tg/lean, indicating a role of MEK6 gene in the metabolism of lipids caused by an HFD’; I don’t agree with the conclusion that this results indicates that MEK6 is involved with lipid metabolism. I do agree that it is having an effect induced by the diet, but it is in my opinion not a direct indicator for changes in lipid metabolism (despite you also observe the changes in eWAT)

  • Figure 5A: next to an HE-staining, the addition of an oil red o staining would add value to the data now presented in the manuscript.

  • As the authors show data of BMDMs isolated from the respective mice, it would be an added value to stain macrophages in the liver and/or WAT to support their claim on inflammation in the in vivo setting (which would make a bridge between their in vitro and in vivo results)

  • The discussion goes very much into detail concerning the mechanistic background of MEK6. Considering the IJMS journal, I understand this choice, but I would advice to balance the discussion a bit clearer with potential application of the described results (see the very final sentence of the discussion on line 373!).

Minor comments:

Line 26: ‘… M2 secretion in BMDM’; I think the authors refer to M2-associated cytokines. Please adapt in such a way

Author Response

We appreciate the Reviewer’s comments. The followings are our point-by-point responses.

Major comments
1. Besides the described effects on lipid metabolism and inflammation, I can imagine that manipulation of MEK6 also has an influence on other metabolic aspects (for instance maybe insulin signaling?). To what extent do the authors believe that the metabolic effects of MEK6 are related to lipids, and do they believe that also other metabolic aspects are involved? (some elaboration on this matter in the discussion of the manuscript would be interesting)
->ANS: We found that MEK6 overexpression decreased UCP2 and UCP1 in the liver and adipose tissue. UCP1 and UCP2 are uncoupling proteins and are known to play an important role in thermogenesis, so they are expected to contribute to the reduction of energy consumption. We did not examine the insulin signaling in MEK6 overexpressed conditions, but the activity of p38 is known to be involved in the regulation of insulin signaling activity. Since MEK6 overexpression regulates p38 activity and cytokines expression, it is expected to be related to insulin signaling, which needs to be confirmed through additional studies.

2. The model the authors use is a ‘full-body’ overexpression of MEK6, making it unclear what the involved of MEK6 is in the respective metabolic organs (liver, heart, WAT). Could the authors elaborate on this matter in the discussion as well? Especially in the context of potential future therapeutics, I think this could have an added value to the paper.
->ANS: RNA expression of the MEK6 gene was expressed in various organs such as the liver, heart, and fat, as the same results in this PCR study. Selected liver and WAT as target organs to confirm the interaction between MEK6 and HFD, we confirmed the lipid metabolism in those organs. In further research studies, we will design the inter-mechanisms study in MEK6 over-expressed organs for potential therapeutics. A paragraph about this content has been added to the discussion, line 356.

3. Line 86: ‘In contrast, MEK6 protein expression in the liver and eWATs was higher in the Non-Tg/DIO than in Non-Tg/lean, indicating a role of MEK6 gene in the metabolism of lipids caused by an HFD’; I don’t agree with the conclusion that this result indicates that MEK6 is involved with lipid metabolism. I do agree that it is having an effect induced by the diet, but it is in my opinion not a direct indicator for changes in lipid metabolism (despite you also observe the changes in eWAT)
->ANS: We intended to comment on the interaction of the MEK6 gene with the high-fat diet, and the content of this text was excluded.

4. Figure 5A: next to a HE-staining, the addition of an oil red o staining would add value to the data now presented in the manuscript.
->ANS: We performed Oil red O staining experiment. Since all lipids such as phospholipids and triglycerides were stained in ORO staining, the difference caused by MEK6 expression was not clear in liver cryosection. Therefore, morphological observations were confirmed through H&E staining using paraffin sections, and lipid accumulation was verified by measuring triglyceride and total cholesterol contents in lipids extracted from the liver.

5. As the authors show data of BMDMs isolated from the respective mice, it would be an added value to stain macrophages in the liver and/or WAT to support their claim on inflammation in the in vivo setting (which would make a bridge between their in vitro and in vivo results) 
-> ANS: Although we extracted BMDM from C57BL/6 mice by 10 per group and conducted the experiment through macrophage cell culture, the number of cells was not sufficient to analysis of inflammatory cytokines. Therefore, we had difficulties in conducting various experiments.

6. The discussion goes very much into detail concerning the mechanistic background of MEK6. Considering the IJMS journal, I understand this choice, but I would advise to balance the discussion a bit clearer with potential application of the described results (see the very final sentence of the discussion on line 373!).
-> ANS: According to the reviewers' valuable comments, our discussion has been clearly modified by clear and final sentence, line 361.

Minor comments:
1. Line 26: ‘… M2 secretion in BMDM’; I think the authors refer to M2-associated cytokines. Please adapt in such a way
-> ANS: Corrected

Thank you so much for insightful comments.

Round 2

Reviewer 1 Report

I have no additional requests